# Progress in the Graphene Oxide-Based Composite Coatings for Anticorrosion of Metal Materials

Shuo Tang [1], Bing Lei [1], Zhiyuan Feng [1], Honglei Guo [1], Ping Zhang [2] and Guozhe Meng [1,*]

1. Marine Corrosion and Protection Team, School of Chemical Engineering and Technology, Sun Yat-sen University, Zhuhai 519082, China; guohonglei@mail.sysu.edu.cn (H.G.)
2. Department of Civil and Environmental Engineering, Faculty of Science and Technology, University of Macau, Taipa 999078, Macau SAR, China
* Correspondence: menggzh3@mail.sysu.edu.cn or mengguozhe@hrbeu.edu.cn

**Abstract:** Graphene oxide (GO), derived from the two-dimensional nanosheet graphene, has received unprecedented attention in the field of metal corrosion protection owing to its excellent barrier performance and various active functional groups. In this review, the protection mechanism "labyrinth effect" of composite coatings against metal corrosion was demonstrated systematically. The origination, structure and properties of GO were also analyzed. Their poor dispersion in polymer and tendency to aggregate as nanofillers in composite coatings are the main limitations during application of the coating fillers. In addition, a comprehensive overview on the perspectives of the surface modification of GO and the multi-functionalization of the composite coatings based on GO were given in particular. Green modification methods, reasonable arrangement of GO sheets in composites and development of multi-functional coatings remain challenges in current studies and should be a focus in the future development of GO-based anticorrosive coatings. This review is of value to researchers interested in the design and application of GO in corrosion protection coatings.

**Keywords:** graphene oxide; two-dimensional fillers; surface modification; multi-functional composite coatings

## 1. Introduction

Currently, with the rapid development of modern industry, metals and alloys have a wide range of applications [1–6]. However, under complex working environments (such as a marine environment), corrosion occurs easily and significantly affects the metal properties. Corrosion degradation is detrimental to metal structure, as severe corrosion shortens the lifetime of the metals and alloys. Failure of metal structure and properties lead to huge economic losses and unexpected disasters [7–9], such as bridges breaking, buildings collapsing, machinery fails and so on. Therefore, corrosion protection of metal/alloys is of great urgency and necessity. Corrosion inhibitor, surface coating and electrochemical protection are typical technologies in metal protection, among which organic coatings are the most commonly used due to their simplicity, high efficiency and cost-effectiveness [10,11].

Various organic coating systems, such as epoxy, polyaniline, polyurethane, alkyd, polymethylmethacrylate, polystyrene, polyamide, polypropylene, polydopamine, silane and polydimethylsiloxane [12–16], are widely used in metal anticorrosion owing to their outstanding property of acting as a physical barrier [17]. However, during application, some microcracks and holes are generated because of the high crosslinking density of the organic coatings, which lead to a certain permeability to $H_2O$, $O_2$ and $Cl^-$ in corrosive media [18,19]. The invasion of these substances obviously affects the anticorrosion performance of the coating and accelerates the corrosion of the metals. Thus, traditional coatings are unable to provide long-term protection. Accordingly, a great deal of work has focused on improving the impermeability of organic coatings to enhance their corrosion resistance. Reinforcements and fillers (e.g., oxide ceramic particles) are supposed to improve the

anticorrosion effect, as they can fill the defects of the coatings to avoid the occurrence of micro-galvanic corrosion [20].

In recent years, the appearance of new two-dimensional fillers opens up a new situation for organic coatings. They can serve as barriers to prolong the propagation path length of the corrosion medium and thus enhance the labyrinth effect [21]. The large aspect ratio and excellent physical barrier make the two-dimensional fillers good candidates for anticorrosion fields. Take steel for example, when exposed to the corrosive environment, the corrosive substance contacts with the metal surface and the metal corrosion occurs gradually. As shown in Figure 1, the corrosion media can easily penetrate the organic coating and then reach the substrate of metal, thus leading to rapid metal corrosion. On the other hand, incorporation of 2D fillers with polymers to make composite coatings provides more winding roads for the penetration of the corrosive substance, which can even block the invasion of harmful ions. Due to their outstanding barrier properties, 2D nanosheets including graphene (Gr), boron nitride (BN), molybdenum disulfide ($MoS_2$), zirconium phosphate (ZrP) and titanium carbide (MXene) are attracting research attention for the applications of metal anticorrosion [22–26]. The corrosion reactions initiated at the metal/coating interface are listed as follows:

$$Fe \rightarrow Fe^{2+} + 2e^- \tag{1}$$

$$Fe^{2+} \rightarrow Fe^{3+} + e^- \tag{2}$$

$$2H_2O + O_2 + 4e^- \rightarrow 4OH^- \tag{3}$$

$$2H_2O + O_2 + 2Fe^{2+} \rightarrow 2H^+ + 2FeOOH \tag{4}$$

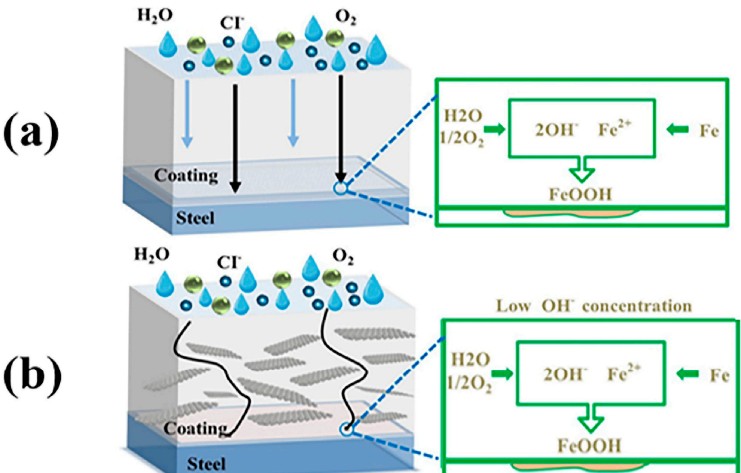

**Figure 1.** Illustration of protective mechanism for: (**a**) coating without 2D fillers, (**b**) coating with 2D fillers [27].

Among these fillers, graphene, a two-dimensional (2D) carbon allotrope, has attracted large attention since its discovery in 2004 [28]. Undoubtedly, the impermeability of graphene plays a critical role in corrosion barrier coatings. The excellent impermeable characters of graphene mainly depend on its atomic structure. The C-C bond length is 0.14 nm and, considering the nuclei of the carbon atoms alone, the pore diameter (or lattice constant) of graphene is only 0.246 nm. Taking the van der Waals radii of carbon atoms (0.11 nm) into account, the pore diameter further decreases to 0.064 nm. Such a small pore size indicates a minimal permeability to corrosive media (such as $H_2O$, $O_2$, $Cl^-$ and $SO_4^{2-}$) [29,30]. Moreover, the dense delocalized electron clouds and free elec-

trons in the π-conjugated carbon networks fill the void within aromatic rings, forming a reactive molecule-repellent field inside the graphene structure [31–33]. Despite the barrier effect, other properties such as thermal stability, excellent processing performance, environmentally friendly, surface flexibility and perfect mechanical properties (Young's modulus reaching 1000 GPa) [34,35] also provide graphene with the possibility for further application in the field of anticorrosion coatings.

Derived from graphene, graphene oxide (GO) has been widely employed in the applications of anticorrosion composite coating. As shown in the Figure 2, according to the data collected by the web of science on the topic of "graphene oxide" and "coating", there are quite a number of articles about this kind of research every year. In recent years, it shows an increasing trend.

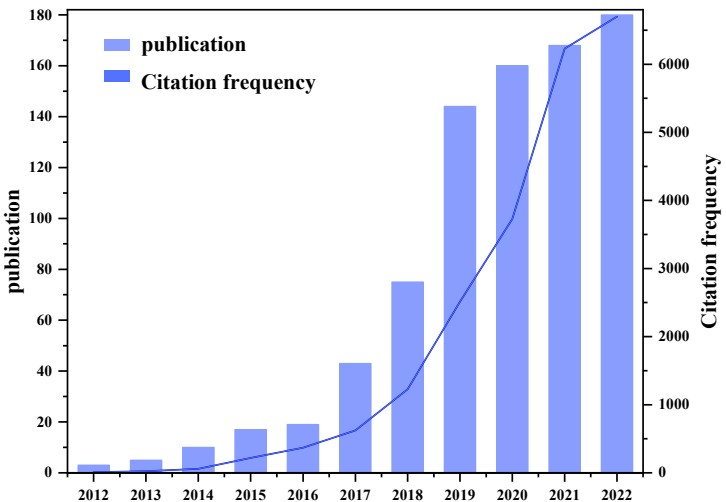

**Figure 2.** The amount of literature on the topic of "graphene oxide" and "coating" in recent years.

In this review, the mechanism of metal corrosion and protection by composite coatings were demonstrated systematically. The origination, the structure and properties of GO were analyzed, and its limitation employed as nanofillers in composite coatings was introduced in detail. In addition, comprehensive perspectives of the surface modification of GO and the multi-functionalization of the composite coatings based on GO were given. Finally, remaining challenges and the future development of GO-based anticorrosive coatings were proposed.

## 2. Graphene Oxide as Composite in Organic Coatings

Although graphene possesses remarkable properties, problems still exist and need to be solved in the practical application. Lei et al. [36] showed that corrosion occurred on the grain boundaries of the graphene layer. The inherent interstitial impurities could lead to a significant reduction in the mechanical strength of graphene and, meanwhile, accelerate the structural transformation under strains. Additionally, the intrinsic vacancy sites benefited the absorption of corrosive substances, which could deteriorate the chemical stability of graphene by accelerating the formation of point defects and promoting substances growth along the grain boundaries [8]. In addition, active functional groups were absent in the graphene structures, making it difficult to form composites with other functional materials. The compatibility between graphene and the organic coatings is another prominent problem of concern. The large aspect ratio and high surface energy of graphene make it easy to obtain agglomeration [37]. Moreover, there are four unpaired electrons in the outermost layer of each carbon atom of graphene, and one of which is free to move within the graphene structure, making it reactive for electron conduction [38]. This unique structure brings graphene excellent electrical conductivity. The conductive

graphene could promote the electrochemical reaction at the interface and thus accelerates the corrosion of metal substrates.

These defects greatly limit the use of graphene in organic coatings. As a result, to overcome these shortcomings and simultaneously take advantages of the merits of the graphene, graphene oxide (GO), a derivative of graphene, aroused wide concern. As a heavily oxygenated form of graphene, graphene oxide is supposed to be the most studied form of graphene and probably causes the most significant changes to graphene application at present [31]. It can be prepared by direct oxidation and exfoliation of graphite. Figure 3 shows the synthesis method of modified Hummer's method, which is one of the most commonly used to produce graphene oxide. After covalent reactions, the surface of graphene is decorated with reactive oxygen functional groups such as carboxyl, epoxy, hydroxyl, carbonyl, phenol, lactone and quinine [39]. During these covalent reactions, a large density of sp$^3$-hybridized carbons in the graphene network are also formed [40], which disrupts the delocalized $\pi$ cloud and thus reduces its electrical conductivity. The transporting speed of charges plays a very important role to corrosion. When applied in practical application, as the service time increases, the metal is inevitably oxidized and thus loses electrons [41]. However, due to the destruction of the conjugated structure, the charge transfer rate on the GO is reduced, which is also beneficial to reducing the rate of electrochemical reaction and thus slow down corrosion. Compared with original graphene, GO becomes more dispersible in water as well as other solvents on account of the rich functional oxygen groups on the surface [42].

To summarize, on one hand, GO retains the excellent characteristics of graphene including shielding effect, high aspect ratio, high specific surface, and an outstanding ability to block the penetration of corrosive substances. On the other hand, the electrical insulativity and the dispersibility of graphene are improved. Most importantly, GO is more easily covalently and/or noncovalently functionalized owing to the presence of functional groups, leading to the improved compatibility between the GO and solvents/polymers [37,43]. GO has been regarded as an ideal nanofiller to prepare composites, and thus has been brought into wide ranges in the anticorrosion field [44,45].

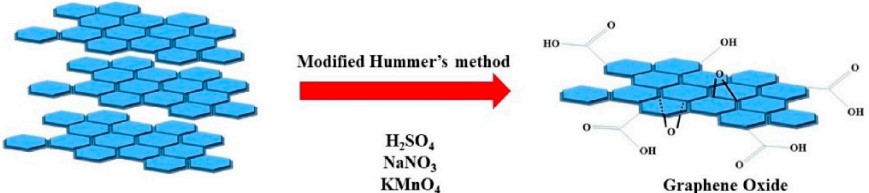

**Figure 3.** Schematic diagram of graphene preparation and structure [46].

Dispersion and compatibility in a polymeric matrix greatly affect the blocking effect of GO. However, similar to the original graphene, the GO sheet has properties of large volume, high surface energy and poor dispersibility in organic coatings [13,27,47]. Due to its high surface energy and intrinsic van der Waals interaction, GO is still easy to aggregate. Once agglomeration happens, the unique aspect ratio of two-dimensional materials disappears, and its barrier properties towards corrosive media will be greatly weakened. As shown in Figure 4, two-dimensional materials are arranged in parallel and stacked layer upon layer in the coating, significantly improving the permeability resistance of the corrosive media in the coating. In other words, this "labyrinth effect" greatly prolongs the diffusion path of the corrosive media and becomes an obstacle to their penetration into the coating. In this way, the corrosion of metal substrate can be effectively delayed. Well-dispersed nanosheets prolong the permeation path of the corrosive materials, but poorly dispersed coating cannot provide effective protection for the metal. The amount of addition also needs to be considered. A too small amount of fillers cannot provide effective protection, while too large an amount of the addition leads to an agglomeration on the contrary.

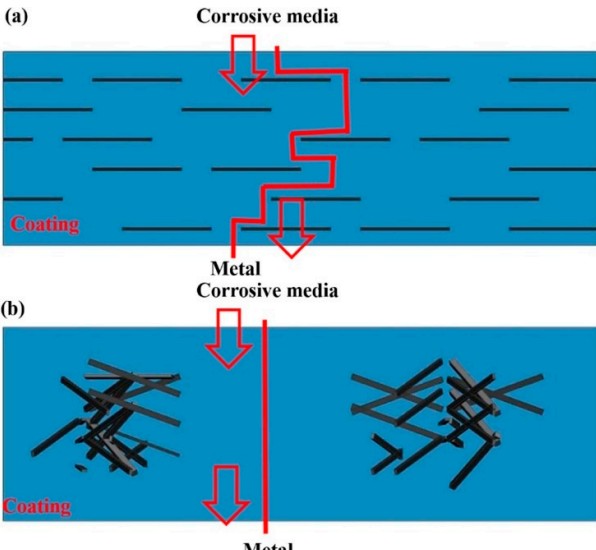

**Figure 4.** (**a**) Well-dispersed graphene prolonging the permeation path of the corrosive materials and (**b**) poorly dispersed coating showing a short permeation path [48].

The density variations at the interface of the polymer and the fillers lead to the formation of microcracks and pores near the fillers, so that the contact between the substrate and the surrounding environment is enhanced. Such cracks and pores may be connected to each other to form rapid channels for the invasion of corrosive materials, leading to localized corrosion spreading under the coating [49,50]. Moreover, differences in physical and chemical properties between filler and polymer result in poor interfacial compatibility and low bonding strength between them [51]. The formed interfaces are sensitive to external stimuli, such as alternating hydrostatic pressure in the deep sea, hydrolysis of electrolytes and erosion of seawater. These external factors will lead to rapid peeling failure of the coating and loss of adhesion in the initial phase. As a result, the diffusion of water through the coating will be accelerated, and its overall corrosion resistance is gradually weakened [52,53].

## 3. Modification of GO

According to the discussion above, the reinforcement of the dispersion and interfacial compatibility between GO and the polymer matrix are the keys to enhance the corrosion resistance of GO-based composite organic coatings. To conquer these obstacles, various studies have been put into the topic based on the optimization of GO nanofiller. Researchers are devoted to modifying the surface of GO to improve its dispersion and compatibility in the composite coating so as to improve the protective effect [54]. Various reagents were applied to change the original properties of the filler surface aiming at forming chemical bonds on the GO/resin interface. By improving the interface bonding ability, the interface compatibility and dispersion of GO can be adjusted and enhanced. According to the types of chemical bonding formed between GO and other materials, the modification methods can be mainly divided into noncovalent bonding modification and covalent bonding modification [55].

### 3.1. Covalent Modification

Due to the existence of abundant oxygen-containing groups such as carboxyl, hydroxyl, epoxide and carbonyl in the structure, it is possible for GO to react with other materials [56]. Covalent bonding modification is mainly achieved by introducing groups to form covalent bonds with GO on the surface by chemical reaction with the existing active groups. Owing to its stability and strong bond energy, covalent bond modification is studied by most researchers. This method is beneficial to maintain the chemical stability and mechanical

properties of GO. Li et al. [57] reported the design and preparation of novel green silk fibroin (SF)-GO nanofillers for a waterborne epoxy coating anticorrosive system. After grafting with SF fibers, GO changed from an inorganic surface to an organic surface. The attached SF fiber could interact with the basic epoxy resin, and thus the dispersibility of the modified GO was greatly improved. Meanwhile, the attached SF fiber on the GO surface could enhance the crosslinking density of the epoxy resins. In the work of Wu et al. [58], aniline oligomer (AO) and grafted GO were used to improve the wear and corrosion performance of phenoxy-resin coating. The grafted AO on the GO surface formed an organic interconnection structure with phenoxy-resin (PHE), which enhanced the interfacial interaction between GO and PHE. After the modification, the composite coatings possessed a better anti-wear and corrosion protection performance. Jiang et al. [59] prepared a polyethyleneimine-grafted graphene oxide (PEI-GO) hybrid material to improve the anticorrosion performance of the waterborne epoxy coating. PEI-GO was uniformly dispersed in the epoxy matrix and the PEI-GO filler exhibited considerable anticorrosive superiorities compared to the primary epoxy coating.

Inhibitors are also used in the graft modification of GO, and the structure of corrosion inhibitors can also improve the protective effect of composite coatings. Chen et al. [60] successfully fabricated a hydrophobic silane/graphene oxide composite coating implanted with benzotriazole (BTAH) inhibitor (BTAH-silane/GO) on copper surface. The formed Si-O-C bond indicated a successful covalent reaction with the silanol groups and a BTAH inhibitor was uniformly embedded to improve the resistance of the composite coating. Liu et al. [61] prepared an ionic liquid–graphene oxide hybrid nanomaterial via a facile grafting reaction between imidazole ionic liquid and GO. Tests proved that the enhanced protective performance of the composite coatings was attributed to the synergistic effect of the impermeable property of graphene nanosheets and the inhibitory function of the imidazole-based ionic liquid. Phytic acid (PA), a green corrosion inhibitor derived from nature, was also employed to modify GO by Wang et al. [62]. The as-prepared PA modified GO (PAGO) thus possessed the passive barrier property and active corrosion inhibition function simultaneously.

In addition to the modified reagents mentioned above, coupling agents are also used to achieve functional modification in practical application. The coupling agents serve as a bridge to combine the polymer and GO. Due to the hydrophilicity of GO, the hydrophilic groups of the coupling agent are connected with GO, and the other groups react with the polar polymer. After modification by coupling agent, long polymer chains are grafted on the edges or surfaces of GO, which can enhance the dispersion and compatibility of GO inside the polymer matrix. At the same time, the modified GO possesses the properties of a polymer, and the inherent defects can be filled by the high crosslinking degree of the polymer, thus improving the overall performance of composite coating.

Liu et al. [63] synthesized novel self-thixotropic hydrogenated castor oil modifying graphene oxide (HMGO) nanosheets from 3-aminopropyltriethoxysilane (APTES) modified GO sheets by grafting the hydrogenated castor oil (HCO) with long alkyl chain molecular structure. HMGO could form a three-dimensional hydrogen network to separate and stabilize the nanosheets in the polyaspartic coating matrix homogeneously. The superiority of the as-prepared composite coating was mainly owing to the fact that the self-thixotropic effect of HMGO met the demands of homogeneous distribution and high compatibility of HMGO nanosheets in the polyaspartic polymer matrix. Pu et al. [64] synthesized the waterborne polyurethane modified by epoxy resin (WEPU) and GO derivatives by copolymerization. The isophorone diisocyanate (IPDI) was grafted to the surface of GO to form the functionalized GO (iGO). Owing to the strong interaction between iGO and WEPU, iGO was then uniformly dispersed in the WEPU emulsions by a prepolymer method and acted as a nanofiller for the composite coating. Ramezanzadeha et al. [65] applied polyaniline (PANI) to functionalize GO and prepared GO-PANI/epoxy coatings. In their design, the titanate coupling agent was used to disperse functionalized GO in a poly(urethane-acrylate) matrix. The titanate coupling agent acted as a bridge to covalently

connect GO and the polymer matrix, and, consequently, increased the compatibility and interactions between GO and the polymer matrix. Through this method, high-performance nanocomposite coatings could be developed [66]. In other work, Zhou et al. [67] used a co-crosslinking strategy between polydopamine (PDA) and polyethyleneimine (PEI) to modify GO. Through the Michael addition and Schiff base reactions, high-density amino-branched PEI and highly adhesive PDA formed covalent bond crosslinks with GO (PDA/PEI-GO). The lamellar spacing of GO was increased by the formed covalently crosslinked network, further enhancing the interfacial bonds between the PDA/PEI-GO lamellae and EP matrix. Due to the improved dispersion and surface activity of GO, the PDA/PEI-GO nanofiller composite coating displayed excellent resistance.

After the successful covalent modification and the introduction of new functional groups, the nonpolarity of modified GO can be improved effectively. As shown in Figure 5, obviously, the storage stability of GO derivatives dispersed in organic matrix solution was improved dramatically compared to the bare GO. This result illustrated that the dispersion and compatibility of the modified GO were significantly enhanced. The improvement of compatibility with polymer further enhanced the protective effect of composite coating.

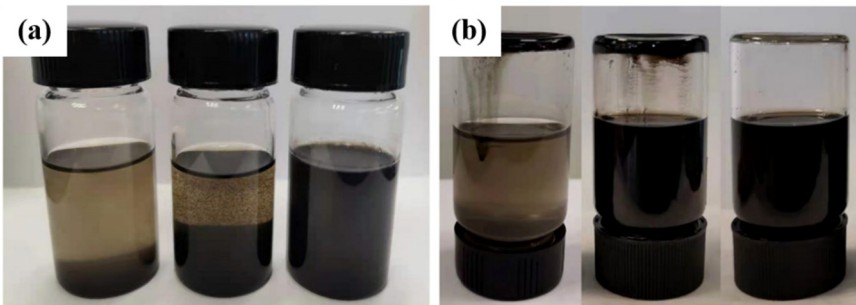

**Figure 5.** Storage stability of GO, MGO and HMGO (from left to right) dispersed in organic matrix solution: (**a**) after being placed 48 h and (**b**) GO and MGO at the bottom of the bottle [63].

### 3.2. Non-Covalent Modification

The noncovalent functional modification is mainly achieved by weak non-bond interaction including π-π bond, hydrogen bond, ionic bond and electrostatic interaction. The noncovalent bond modification of GO can retain the original structure and excellent properties of GO and increases the distance between GO nanosheet layers, thereby reducing agglomeration and improving the dispersion of GO. Without the usage of a large amount of organic solvents, noncovalent modification is more environmentally friendly compared to covalent modification.

For example, Gao et al. [68] fabricated amorphous cellulose (AC) edge-functionalized GO by electrostatic self-assembly of negatively charged AC (NAC) and positively charged GO (NGO). The schematic diagram of preparation progress is shown in Figure 6. The formed NAC enhanced the anticorrosion performance of NGO as a nanofiller in waterborne epoxy resin (WEP) coatings, due to the well dispersion of NAC/NGO composite and its strong interfacial interactions with the matrix.

Amrollahi et al. [69] developed a polyaniline (PANI)-modified GO for obtaining a high-performance epoxy nanocomposite film with anticorrosion properties. Analysis demonstrated that the successful polymerization of the emeraldine base form of polyaniline on the GO surface could be ascribed to two forms of noncovalent bonding. One was the π-π interactions between the quinoid ring of the PANI and the basal plane of GO, and the other was the covalent bonding through reaction with epoxide groups. The as-developed composite coating also showed better dispersion of GO modified with polyaniline. Wu et al. [70] chose the hydrophilic GO as an intercalator to exfoliated hexagonal boron nitride (h-BN) and enhanced its dispersion in water-borne epoxy (WBE) directly. Similarly, its π-π interaction between GO and h-BN enabled h-BN homogeneously to load on the surface of GO. Performance of the GO/h-BN/WBE composite coatings showed prominent anticorro-

sion properties due to the homogeneous dispersion of GO/h-BN (1:1 *w/w*) composite as well as the barrier effect of GO and h-BN for the corrosive medium.

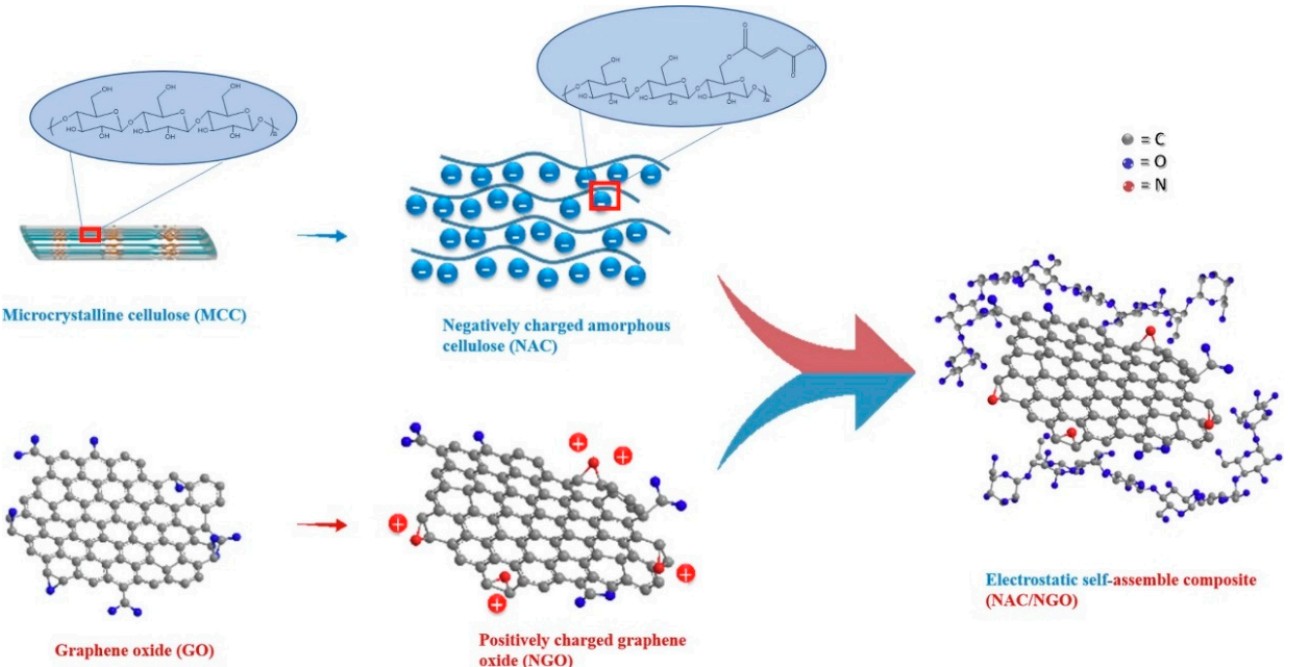

**Figure 6.** The preparation of edge-functionalized graphene oxide by amorphous cellulose (NAC/NGO) [68].

Non-covalent modification effectively inhibits the agglomeration of GO by increasing the layer spacing. The dispersion and compatibility of GO in polymer can also be improved with appropriate noncovalent modification. As shown in Figure 7, after noncovalent modification, the GO composite showed a homogeneous dispersion state under the optimal ratio. Despite the characteristic of maintaining the original structure and high specific surface area of GO [71], the noncovalent modification is generally applied as a supplement to covalent modifications, because of the weaker interaction between the GO and the functional groups and less stability of the coating material. The sole application of noncovalent modification is relatively limited.

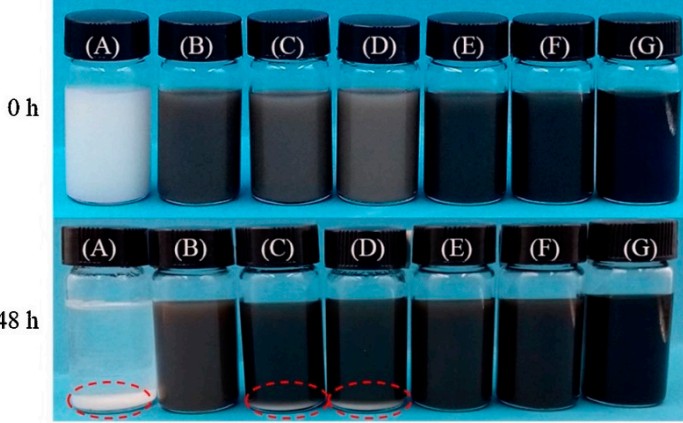

**Figure 7.** Dispersion state of the: (**A**) h-BN, (**B**) GO/h-BN (1:1 *w/w*), (**C**) GO/h-BN (1:2 *w/w*), (**D**) GO/h-BN (1:3 *w/w*), (**E**) GO/h-BN (2:1 *w/w*), (**F**) GO/h-BN (3:1 *w/w*) and (**G**) GO in water 48 h after ultrasonication for 30 min [70].

As is shown in Figure 8, the low-frequency impedance modulus ($|Z|_{0.01Hz}$) of the composite coatings after GO modification was compared with that of the blank samples. Obviously, after the covalent/noncovalent modification, the dispersion and compatibility of nanofillers were enhanced, leading to much better performance of composite coatings on metal protection.

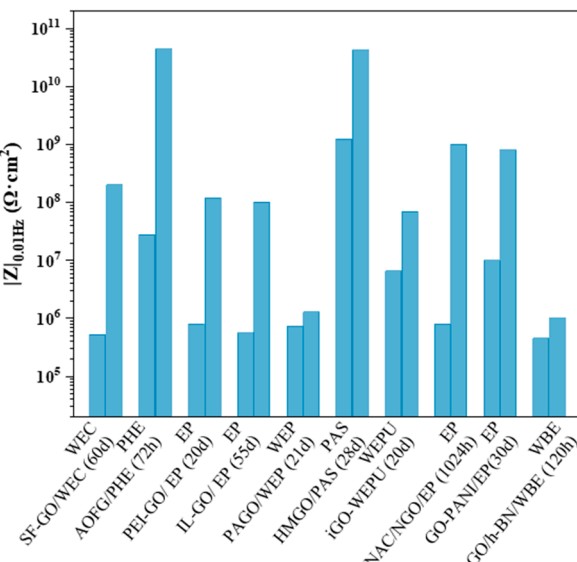

**Figure 8.** Diagram of low-frequency impedance modulus ($|Z|_{0.01Hz}$) of composite coating and blank sample under optimal conditions reported in the literature listed above.

## 4. Graphene Oxide-Based Multi-Functionalization Coatings

Generally, the traditional composite coatings focus more on the long-term performance, durability as well as reliability for metal protection. However, when facing a complex service environment, the multi-functional anticorrosion coatings gradually attract more and more attention. These multi-functionalization anticorrosion coatings that are self-sensing, self-healing, wear-resisting, antibiosis and super-hydrophobic have intensively been developed to promote advanced applications [72]. To meet the demands of the multi-function in composite coatings, GO plays a significant role due to its excellent properties discussed above. Instead of focusing on its dispersion and compatibility, this part discusses several kinds of GO-based multi-functionalization coatings.

### 4.1. Self-Healing Composite Coatings

As discussed above, the presence of microcracks in polymer coatings degrades the structural stability and lifespan of the protective layers. In general, once the anticorrosive coating is applied and fully cured on the substrate, it is difficult to inspect the coating for damage and repair it in a timely manner. As a result, accomplishing coatings with self-healing properties has attracted considerable attention for polymer systems. The strategy of self-healing is mainly based on a responsive approach to repair material damage during its service life. External environment, such as vibration, pressure, pH value, heat, light and humidity can stimulate active response to change. Corrosion inhibitors can protect the metal by delaying or even inhibiting the corrosion. However, the direct addition of corrosion inhibitors to coating formulations can cause a deactivation of the inhibitor and/or fast degradation of adhesion and barrier properties of the coating [73,74]. To solve these issues, micro- or nanoparticles encapsulating/loading inhibitors were embedded in coatings [75–78].

GO, with outstanding barrier performance and rich surface groups, can also serve as a container of corrosion inhibitors in the field of intelligent self-repair. Microcapsules with reasonable mechanical stability, thin wall thickness and high loading capacity act

as permeation barriers to prevent not only the diffusion of corrosive medium but also the leakage/solidification of healing agents. On this occasion, GO with outstanding impermeability, high specific surface area and abundant surface functional groups can be fabricated not only as platforms for stimuli-responsive nanocontainers but also as fillers to block corrosive components. Corrosion inhibitors can be entrapped in its capsules, and the existing pores of the container allow intelligent release by different triggering factors [79].

As is shown in Figure 9, the containers with inhibitors are well dispersed in the polymer matrix to enhance the "labyrinth effect" when the composite coatings are complete, which prolongs the path of corrosive media or even blocks their invasion to the substrate. At this stage, the containers serve as the physical barrier to improve the corrosion resistance of composite coating. On the other hand, once the damage of coatings occurs, corrosive substances from the environment penetrate the coating and reach the metal substrate. Thus, the microcurrent is easily generated in the microscopic areas and the micro-corrosion happens without any hesitation. Straightaway, a "stimulus signal" prompts inhibitors to be released, and the released inhibitors will be adsorbed on the metal surface, exerting their anticorrosion effect.

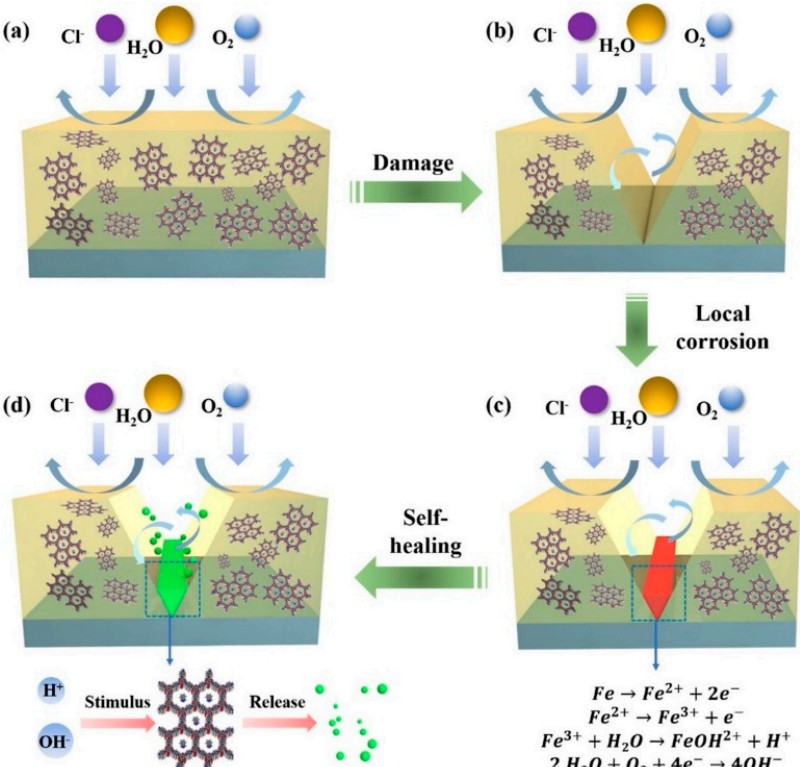

**Figure 9.** Anti-corrosion and self-repair mechanism of coatings: (**a**) complete coating, (**b**) damaged coating, (**c**) local corrosion occurs and (**d**) coating self-healing [80].

Zhou et al. [81] reported the successful fabrication of a novel dual self-healing anticorrosion coating based on benzotriazole loaded $TiO_2$ nanocapsule (BTC) modified GO sheets and the multibranched waterborne polyurethane (WPU). GO sheets were first modified with amino groups (GO-$NH_2$) and then grafted with 3-isocyanatopropyltriethoxysilane (GNI) to further react with the BTCs (GNI-BTCs) as shown in Figure 10. After a series of reactions, BTCs were well dispersed on the surface of GNI sheets to improve the dispersion of GNI into the waterborne coatings.

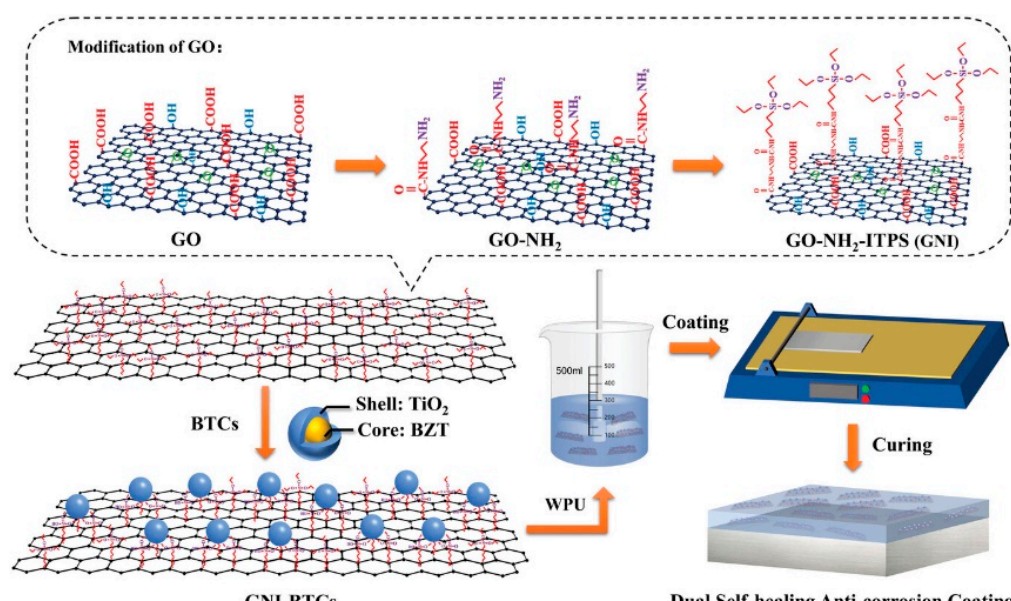

**Figure 10.** Fabrication of the dual self-healing anticorrosion coating [81].

Luo et al. [82] used isophorone diisocyanate, polytetramethylene ether glycol, dimethylglyoxime and glycerol to prepare polyurethane. GO was added to obtain a polyurethane/GO composite with self-healing and shape memory properties. Yu et al. [83] fabricated GO microcapsules containing light-curing epoxy resin based on the Pickering emulsions in a single step. The chemical stability of microcapsules was improved through chemical stitching of GO nanosheets with polyether amine. A self-assembly process was employed by Li et al. [84] to prepare the graphene oxide microcapsules (GOMCs) containing linseed oil as the healing agent. The nanometer-thick shells of GOMCs were built by the liquid crystalline assembling of GO sheets, forming at the liquid–liquid interface in Pickering emulsions. The as-prepared composite coatings not only possessed self-healing properties, but also showed excellent anticorrosion properties based on the physical barrier property of the GO shell. Chen et al. [85] developed a new strategy to improve anticorrosion performance of epoxy coatings, in which halloysite nanotubes (HNTs) were combined with GO. On one hand, the corrosion inhibitors loaded in nanocontainers would be released when the coating was damaged and subsequently prevented metal from further corrosion because of the pH-responsive ability of the nanocomposite. On the other hand, GO exerted the physical barrier property to protect the metal matrix against a corrosive medium.

In all, in order to avoid the direct addition of corrosion inhibitors that could be harmful to the polymer matrix, the GO-based containers provide a buffer for the release of an inhibitor and achieve long-term protection. When corrosion occurs, the GO-based containers can release corrosion inhibitor and repair the corroded area in time, realizing the self-healing effect.

### 4.2. Self-Warning Composite Coatings

Generally speaking, microcracks that are not easy to detect are the initial stage of various properties degradation of polymer materials. Lacking timely detection and repairment of these defects, corrosion deterioration may develop rapidly, resulting in rapid degradation of the protective coatings for substrate matrix. Therefore, the self-warning ability of anticorrosive coating materials is of great significance in practical applications. Timely sensing of the coating damage leads to proper healing and maintenance procedures to improve structural integrity and avoid unexpected failure [86]. It is of great significance to provide autonomous early warning by detecting the corrosion reaction at an early stage. Only in this way, the timely maintenance can be conducted. Although urgent as it is, this kind of research is still in the preliminary stage and needs more investigation.

Metal corrosion usually involves two reactions. Anodic reaction is metal electron loss oxidation to generate related metal ions, which may be accompanied by hydrolysis and acidification of metal ions according to environmental conditions. Cathode reaction includes electron reduction of oxygen or water in etched areas to form hydroxide ions [87]. Once corrosion occurs, a change in the concentration of metal ions is inevitably generated, while possible acidification due to the formation of hydroxide ions also results in a change in the pH of the etched area. Based on the ions and pH changes produced by corrosion, we can choose appropriate indicators for warning.

Ionic indicators are related to the detection of various metal ions. For example, sulfosalicylic acid and potassium thiocyanate are indicators of ferric ion; rhodaminohydrazine and phenanthroline are indicators of ferrous ions; 8-hydroxyquinoline and coumarin have a fluorescence response to aluminum ion; rhodamine ethylenediamine is an indicator of copper ions. Common pH indicators are sensitive to pH change within a given pH range. Here, we listed several pH-based indicators that can be used for corrosion monitoring in Table 1.

**Table 1.** Some pH-sensing molecules that have the potential for corrosion detection.

| Sensing Molecules | pKa | Transition pH Range |
| --- | --- | --- |
| Phenolphthalein | $OH^-$ | 8.2–10.0 |
| Bromocresol green | $OH^-$ | 8.0–10.7 |
| Cresol red | $OH^-$ | 7.2–8.8 |
| Methyl red | $H^+$ | 4.2–6.2 |
| Bromothymol blue | $H^+$ | 6.0–7.6 |

Similarly, it is not suitable to add the indicator directly to the coating, which may lead to their early leakage and properties reduction in the polymer matrix. In the study of Tiago et al. [88], the coating with direct addition of phenolphthalein performed even worse than that of pure coating samples. As their work presented, the uncoated phenolphthalein reduced the crosslinking degree of resin during curing. On the contrary, with the protection of the silica nanocapsules shell, detrimental interaction with the active compound in the coating could be minimized, which proved the importance of encapsulation of phenolphthalein by the container.

As a result, combined with GO, a novel design of corrosion alarming coating was proposed. Take for instance, 1,10-phenanthroline (Phen) was employed as the corrosion indicator, which was grafted on GO surfaces so that the Phen could distribute uniformly in the polymer matrix. In the work of Li et al. [89], they prepared a self-sensing polymer composite coating with functionalized GO. The GO was chemically modified with Phen which could form a red complex with $Fe^{2+}$ to realize a corrosion alarm at an early stage. At the same time, the addition of laponite RD (laponite) improved the dispersion of Phen-modified GO in the polyurethanes (PU) matrix. Results illustrated that the as-prepared composite coatings showed higher corrosion resistance compared to the pure PU coatings.

*4.3. Other Multi-Functional Coatings*

In addition to excellent barrier properties, its large surface area, rich surface functional groups and good mechanical properties also promote the GO application in coating fields of super hydrophobic, antibacterial, wear resistance and so on. GO with abundant oxygen-containing functional groups and nanoscale size was prepared and incorporated into waterborne polyurethanes (WPU) by chemical grafting to improve the dispersion in WPU [26], resulting in excellent mechanical properties and solvent resistance of the coating. Besides the good compatibility with WPU, the tensile strength of that coating film increased by 64.89%, and the abrasion resistance and pendulum hardness increased by 28.19% and 15.87%, respectively. This chemical grafting strategy provides a feasible way to improve the dispersion of GO in coatings and of reference value in the modification of waterborne coatings.

Superhydrophobic silane/GO composite coating, with excellent anticorrosion performance and durability, was also successfully synthesized on copper surface using simple dipping and subsequent curing procedure [90]. The prepared superhydrophobic silane/GO composite coating possessed the largest impedance modulus and the protection efficiency was over 99% after exposed to 1 M NaCl solution for 120 h. Apart from these applications listed above, studies on the nanotribological properties of GO indicated that excellent lubrication performance and wear resistance of GO made it a potential high-performance nano-lubricating material [91,92]. Zhang et al. [93] mixed GO with WC-17Co alloy powder. GO was embedded in the coating as transparent thin sheets. The friction coefficient of the GO coating was reduced by approximately 22% compared to that of the original coating. The formation of lubrication films in the micro-area improved the self-lubrication and anti-wear effects of the coatings. Similarly, GO was incorporated into WC-12Co powder via wet ball milling and spray granulation, and GO was embedded in the structure in a transparent and thin-layer state [94]. Compared to the friction coefficient (0.6) of the WC-12Co coating obtained at room temperature, the friction coefficient of the GO/WC-12Co coating was decreased by approximately 50%. In another work [95], the NiCr-WC-Al$_2$O$_3$ composites with the addition of GO were fabricated by powder metallurgy technique. Results showed that in high temperature ranges, the composites with the addition of 3 wt% GO exhibited the lowest friction coefficient (0.41) and wear rate ($1.0 \times 10^{-5}$ mm$^3$/Nm) at 700 °C. In the above studies, the improvement of GO nanosheets to the tribological performance of metal is mainly due to excellent mechanical and tribological properties. In addition, GO is thought to be a promising antibacterial material. The oxygen-containing functional groups endowed GO with good hydrophilicity, dispersity and biocompatibility, making GO a promising biomedical applications candidate [96–98]. A number of studies have proved the strong antibacterial activities of GO, and its antibacterial activity is considered to be mediated by the physicochemical interaction between GO and microbes [99,100].

In all, the multi-functional composite coatings refer to the materials with practical functions including self-healing, self-warning, superhydrophobic, wear-resistant, antibacterial, etc. Apart from enhancing the barrier property of coating as reinforcements/fillers, more efforts should be put into the study of multi-functional coatings based on the excellent properties of GO.

## 5. Conclusions and Perspective

Graphene oxide is distinct in the field of organic composite coatings in corrosion protection due to its outstanding barrier properties. At present, research is mainly devoted to two aspects: one is to improve the dispersion and compatibility of GO in polymer through modification, so as to further improve the barrier performance and the protection effect of organic composite coating. The other is based on the structure of GO and abundant covalent bonds on it to carry out multi-functionalization of organic coatings such as self-sensing, self-healing and wear-resist and antibacterial so as to give more possibilities to composite coatings.

Short as the history of GO to be a filler of organic composite coating is, the study of its reaction mechanism is limited. More work is needed to be done in the future.

1.  The stable effect and strong interaction between GO and functional groups make the covalent modification the mainstream method of GO. Nevertheless, this method destroys the original structure of GO to some extent. During the reactions, a large amount of highly toxic organic reagents are used. Therefore, more systematic and comprehensive investigation is needed to develop green modification methods that can preserve the original structure and properties of graphene, so as to ensure dispersion while reducing risk of environmental pollution and health hazards.

2.  In addition to the problems of dispersion and compatibility highlighted above, the arrangement of GO and its derivatives in polymers also make great sense on its barrier effect. Due to the high aspect ratio of GO nanomaterials, the relative arrangement of corrosive media and nanomaterials (parallel or perpendicular to the path of corrosive

media) has a great difference in exerting its labyrinth effect during the process of penetrating coating. In this regard, to realize orderly and reasonable arrangement of GO sheets in polymer composites is still a big challenge for the fabrication of multi-functional and high-performance composites. The orientation arrangement of such two-dimensional lamellar materials induced by external magnetic field or electric field are worth studying.

3. Compared with the widely employed epoxy resins, the content of volatile organic compounds (VOCs) in water-based epoxy resins is much lower, which caters to people's demands for green chemistry and sustainable development. However, low hardness, long curing time and insufficient shielding effect on corrosive media make water-based epoxy resin unable to replace the application of epoxy resins. Study of ideal polymers should still be promoted.

4. The development of multi-functional coatings (such as self-sensing and self-healing), especially in special environments such as deep sea, is still in the initial stage. Additionally, the trigger conditions of "intelligent" are relatively strict and insensitive, making it far from ready for industrial employment. A lot of work is needed to make improvement on it.

**Author Contributions:** S.T.: conceptualization, methodology, software, investigation, formal analysis, writing—original draft; B.L.: data curation, writing—original draft; Z.F.: visualization, investigation; H.G.: resources, supervision; P.Z.: software, validation; G.M. (corresponding author): conceptualization, funding acquisition, resources, supervision, writing—review and editing. All authors have read and agreed to the published version of the manuscript.

**Funding:** This research was funded by the National Natural Science Foundation of China grant number [Nos. 52171093, 51,771,061 and U20A20233].

**Institutional Review Board Statement:** Not applicable.

**Informed Consent Statement:** Not applicable.

**Data Availability Statement:** Not applicable.

**Conflicts of Interest:** The authors declare no conflict of interest.

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
