# Peer review of "Progress in the Graphene Oxide-Based Composite Coatings for Anticorrosion of Metal Materials"

_coatings, doi:10.3390/coatings13061120_

Round 1

Reviewer 1 Report

Comments from the reviewer for the manuscript number 2451286 entitled

Comments for the authors

The manuscript number 2451286 entitled “Progress in the Graphene Oxide-Based Composite Coatings for Anticorrosion of Metal Materials” by Shuo Tang et al. It is a very good work and well written. However, few minor English mistake are there in the manuscript. So it can be accepted after incorporating the minor corrections, which are given below.

Needed minor corrections

Abstract

1.      All right.

Keywords

2.      All right.

1. Introduction

3.      Line 27. ---a wide range of applications[1–6].

A gap is necessary before citation of references. So change as

----a wide range of applications [1–6].

Similar changes are necessary in lines 30, 35, 38, 39, 42, 48, 51, 62, 68, 77, 80, 98, 104, 108, 110, 118, 120, 122, 123, 125, 130, 138, 143, 164, 166, 171, 178, 184, 188, 192, 198, 213, 218, 237, 248, 282, 289, 304, 322, 337, 339, 348, 361, 370, 372, 381, 400, 407,  454, 461, 463, in Figure 1, Figure 4, Figure 6, Figure 9, Figure 10 captions as well as other part of text.

2. Graphene oxide as composite in organic coatings

4.      Line 102.--- of corrosive substances which could deteriorate—

A comma is necessary before “which”. So change as

---of corrosive substances, which could deteriorate---

5.      Line 120. ---GO which is still---

A comma is necessary before “which”. So change as

---GO, which is still---

3. Modification of GO

3.1. Covalent modification

6.      All right.

3.2. Non-covalent modification

7.      All right.

.

4. Graphene oxide-based multi-functionalization coatings

4.1. Self-healing composite coatings

8.      All right.

4.2. Self-warning composite coatings

9.      All right.

4.3. Other multi-functional coatings

10.  All right.

5. Conclusion and perspective

11.  ---can be found at? The above sentence is not complete.

Figures

12.  All right.

References

13.  Why the references number have been given twice?

Author Response

Dear Editor

We thank Editor and Reviewers for allowing us to revise our manuscript, and we appreciate the editor and reviewers for their constructive comments and suggestions on our manuscript.

We have studied the reviewer’s comments carefully and have made revisions which are marked in yellow on the paper. We have tried our best to revise our manuscript according to the comments. Please find the revised version attached, which we would like to submit for your kind consideration.

We would like to express our great appreciation to you and the reviewers for the comments on our paper. Looking forward to hearing from you. 

All the best

Guozhe Meng

-------------------------------------------------------------------------------

Reviewer1:

Dear Prof./Dr.:

We would like to thank the Reviewer for his/her thoughtful comments and efforts toward improving our manuscript.

Comments for the authors

The manuscript number 2451286 entitled “Progress in the Graphene Oxide-Based Composite Coatings for Anticorrosion of Metal Materials” by Shuo Tang et al. It is a very good work and well written. However, few minor English mistake are there in the manuscript. So it can be accepted after incorporating the minor corrections, which are given below.

Needed minor corrections

Response: Thanks for your kind suggestions. We have studied the reviewer’s comments carefully and have made revisions which are marked in yellow in the revised paper

Abstract

  1. All right.

Keywords

  1. All right.
  2. Introduction
  3. Line 27. ---a wide range of applications[1–6].

A gap is necessary before citation of references. So change as

----a wide range of applications [1–6].

Similar changes are necessary in lines 30, 35, 38, 39, 42, 48, 51, 62, 68, 77, 80, 98, 104, 108, 110, 118, 120, 122, 123, 125, 130, 138, 143, 164, 166, 171, 178, 184, 188, 192, 198, 213, 218, 237, 248, 282, 289, 304, 322, 337, 339, 348, 361, 370, 372, 381, 400, 407,  454, 461, 463, in Figure 1, Figure 4, Figure 6, Figure 9, Figure 10 captions as well as other part of text.

-Response: Thanks for your kind suggestions, we have revised it accordingly.

  1. Graphene oxide as composite in organic coatings
  2. Line 102.--- of corrosive substances which could deteriorate—

A comma is necessary before “which”. So change as

---of corrosive substances, which could deteriorate---

-Response: Thanks for your kind suggestions, we have revised it accordingly.

  1. Line 120. ---GO which is still---

A comma is necessary before “which”. So change as

---GO, which is still---

-Response: Thanks for your kind suggestions, we have revised it accordingly.

  1. Modification of GO

3.1. Covalent modification

  1. All right.

3.2. Non-covalent modification

  1. All right.
  2. Graphene oxide-based multi-functionalization coatings

4.1. Self-healing composite coatings

  1. All right. 

4.2. Self-warning composite coatings

  1. All right. 

4.3. Other multi-functional coatings

  1. All right.
  2. Conclusion and perspective
  3. ---can be found at? The above sentence is not complete. 

-Response: Thanks for your kind suggestions, we have checked it again and revised it carefully.

Figures

  1. All right. 

References

  1. Why the references number have been given twice? 

-Response: Thanks for your kind suggestions, we have uploaded it accordingly.

Reviewer 2 Report

 The Article “Progress in the Graphene Oxide-Based Composite Coatings for Anticorrosion of Metal Materials” by Guozhe Meng is devoted to reviewing the latest research in the application of graphene oxide and its modifications to composite anticorrosion materials. The review reveals the main aspects and will be of interest to researchers specializing in the field of materials science and corrosion protection.

The main comments of the reviewer are given below.

Poor quality of figures 1 and 6. Gray signatures are almost invisible. In figure 6, the captions should be enlarged, the text is hard to read.

The level of the English text should be improved. Sometimes authors use somewhat unscientific expressions, such as "came into people's eyes", "Bearing this in mind", "Thanks to", etc. There are frequent repetitions in sentences (for example, in Abstract “This review is interesting study to researchers interested in the design and application of GO in corrosion protection coatings.”)

Row 119. It is worth mentioning Brodie, equally with Hammers, as the founder of the synthesis of GO. In addition, the primary source for graphene oxide is still not graphene, but graphite.

Rows 122, 123 “During reactions, a large density of sp3 -hybridized carbons in the graphene network are also formed[42], which disrupts the delocalized π cloud thus reducing its electrical conductivity”. It is not clear from the text which reactions are meant - the oxidation of graphite, or chemical modifications of graphene oxide; or the sentences should be reformulated.

Row 124. "During practical application, as the service time increases, the metal is inevitably oxidized and thus loses electrons[43]". The sentence does not continue the previous paragraph. Reference 43 does not match the text. The list of references should be corrected: duplication of numbers should be removed.

Line 143. “poor solubility in organic coatings” Graphene oxide is not able to dissolve, but only disperse in various media.

The level of the English text should be improved. Sometimes authors use somewhat unscientific expressions, such as "came into people's eyes", "Bearing this in mind", "Thanks to", etc. There are frequent repetitions in sentences (for example, in Abstract “This review is interesting study to researchers interested in the design and application of GO in corrosion protection coatings.”)

Author Response

Dear Editor

We thank Editor and Reviewers for allowing us to revise our manuscript, and we appreciate the editor and reviewers for their constructive comments and suggestions on our manuscript.

We have studied the reviewer’s comments carefully and have made revisions which are marked in yellow on the paper. We have tried our best to revise our manuscript according to the comments. Please find the revised version attached, which we would like to submit for your kind consideration.

We would like to express our great appreciation to you and the reviewers for the comments on our paper. Looking forward to hearing from you. 

All the best

Guozhe Meng

-------------------------------------------------------------------------------

Dear Prof./Dr.:

We would like to thank the Reviewer for his/her thoughtful comments and efforts toward improving our manuscript.

 The Article “Progress in the Graphene Oxide-Based Composite Coatings for Anticorrosion of Metal Materials” by Guozhe Meng is devoted to reviewing the latest research in the application of graphene oxide and its modifications to composite anticorrosion materials. The review reveals the main aspects and will be of interest to researchers specializing in the field of materials science and corrosion protection.

Response: Thanks for your kind suggestions. We have studied the reviewer’s comments carefully and have made revisions which are marked in yellow in the revised paper

The main comments of the reviewer are given below. 

Poor quality of figures 1 and 6. Gray signatures are almost invisible. In figure 6, the captions should be enlarged, the text is hard to read. 

-Response: Thanks for your kind suggestions, we have tried our best to provide the high-resolution figures. However, figure 1 was downloaded directedly from the website of the paper and the original picture was so clear. We have enlarged the captions in figure 6 to make it easier to read.

The level of the English text should be improved. Sometimes authors use somewhat unscientific expressions, such as "came into people's eyes", "Bearing this in mind", "Thanks to", etc. There are frequent repetitions in sentences (for example, in Abstract “This review is interesting study to researchers interested in the design and application of GO in corrosion protection coatings.”)

-Response: Thanks for your kind suggestions. We have revised and deleted these unfit expressions, some other language changes have also been made to improve our expressions. (line 117, 129, 142,149,246,424)

Row 119. It is worth mentioning Brodie, equally with Hammers, as the founder of the synthesis of GO. In addition, the primary source for graphene oxide is still not graphene, but graphite.

-Response: Sorry for the wrong information, we have carefully check it again and deleted the wrong information. We agree with your opinion that the method of Brodie is also important for the synthesis of GO, and the modified Hummer’s method is also one of the most commonly used method in the laboratory to produce graphene oxide. Both graphene and graphite can be used to synthesize GO. We have revised there to be “Figure 3 shows the synthesis method of modified Hummer’s method, which is one of the most commonly used one to produce graphene oxide.”

Rows 122, 123 “During reactions, a large density of sp3 -hybridized carbons in the graphene network are also formed[42], which disrupts the delocalized π cloud thus reducing its electrical conductivity”. It is not clear from the text which reactions are meant - the oxidation of graphite, or chemical modifications of graphene oxide; or the sentences should be reformulated.

-Response: Thanks for your kind suggestions. In the process of preparing graphene oxide from graphene, a series of covalent reactions occur. These covalent reactions lead to changes in two aspects, the decoration of reactive oxygen functional groups and the disruption of delocalized π cloud. The latter one results in a decrease in electrical conductivity. We have revised these expressions to “After covalent reactions, the surface of graphene is decorated with reactive oxygen functional groups such as carboxyl, epoxy, hydroxyl, carbonyl, phenol, lactone, and quinine [41]. During these covalent reactions, a large density of sp3-hybridized carbons in the graphene network are also formed [42], which disrupts the delocalized π cloud and thus reduces its electrical conductivity. The speed of charges transporting plays a very important role to corrosion. When applied in practical application, as the service time increases, the metal is inevitably oxidized and thus loses electrons [43]. However, due to the destroy of the conjugated structure, the charge transfer rate on the GO is reduced, which is also beneficial to reducing the rate of electrochemical reaction and thus slow down the corrosion.”

Row 124. "During practical application, as the service time increases, the metal is inevitably oxidized and thus loses electrons[43]". The sentence does not continue the previous paragraph. Reference 43 does not match the text. The list of references should be corrected: duplication of numbers should be removed.

-Response: Thanks for your kind suggestions. Here we have added descriptions that relate conductive properties to corrosion of metal. “The speed of charges transporting plays a very important role to corrosion. When applied in practical application, as the service time increases, the metal is inevitably oxidized and thus loses electrons [43]. However, due to the destroy of the conjugated structure, the charge transfer rate on the GO is reduced, which is also beneficial to reducing the rate of electrochemical reaction and thus slow down the corrosion.” The list of references has been corrected accordingly.

Line 143. “poor solubility in organic coatings” Graphene oxide is not able to dissolve, but only disperse in various media.

-Response: Thanks for your kind suggestions. We have revised here to be “poor dispersibility”.

Comments on the Quality of English Language

The level of the English text should be improved. Sometimes authors use somewhat unscientific expressions, such as "came into people's eyes", "Bearing this in mind", "Thanks to", etc. There are frequent repetitions in sentences (for example, in Abstract “This review is interesting study to researchers interested in the design and application of GO in corrosion protection coatings.”)

-Response: Thanks for your kind suggestions. We have revised and deleted these unfit expressions, some other language changes have also been made to improve our expressions. (line 117, 129, 142,149,246,424)

Reviewer 3 Report

Although it is interesting to conduct a research on the development of GO-based composite coatings in the submitted review study, it is very weak in terms of its application as a coating. For this reason;

Results on the application of GO-based composite coatings with interface development on different substrates should definitely be given with SEM,EDS,etc images.

The journal is 'Coatings' journal so it doesn't seem appropriate for this coating layers as such. For the 'Coatings' journal, the application of improved composite coatings on other substrates should be considered rather than modifying the GO surface.

Studies examining surface coatings and interface characteristics should be included with SEM,EDS,etc images.

The study contains very few figures and tables. studies in the literature should be given in the form of tables in comparison. The results obtained in the literature about surface coatings should be presented as figures.

GO composites are frequently used on different substrates to increase their wear and corrosion resistance. How the coatings applied on the substrates affect the wear and corrosion resistance should be given by comparing the examples in the literature.

Author Response

Dear Editor

We thank Editor and Reviewers for allowing us to revise our manuscript, and we appreciate the editor and reviewers for their constructive comments and suggestions on our manuscript.

We have studied the reviewer’s comments carefully and have made revisions which are marked in yellow on the paper. We have tried our best to revise our manuscript according to the comments. Please find the revised version attached, which we would like to submit for your kind consideration.

We would like to express our great appreciation to you and the reviewers for the comments on our paper. Looking forward to hearing from you. 

All the best

Guozhe Meng

-------------------------------------------------------------------------------

Dear Prof./Dr.:

We would like to thank the Reviewer for his/her thoughtful comments and efforts toward improving our manuscript.

Although it is interesting to conduct a research on the development of GO-based composite coatings in the submitted review study, it is very weak in terms of its application as a coating. For this reason;

Thanks for your kind suggestions. We have studied the reviewer’s comments carefully and have made revisions which are marked in yellow in the revised paper

Results on the application of GO-based composite coatings with interface development on different substrates should definitely be given with SEM,EDS,etc images.

-Response: Thanks for your kind suggestions. We totally agree with your idea that the interface characteristics are very important for the performance of coatings. However, the main focus of the paper based on the discussion of the main limitations of GO application including poor dispersion in polymer and tendency to aggregate as nanofillers in composite coatings. To illustrate these two points, the relate “dispersion state experiment” (figure5 and figure7) can be more intuitively to show the modification of fillers. In our own researches, the interface performance (SEM, EDS, etc) between substrates and coatings will be systematically studied.

 The journal is 'Coatings' journal so it doesn't seem appropriate for this coating layers as such. For the 'Coatings' journal, the application of improved composite coatings on other substrates should be considered rather than modifying the GO surface.

-Response: Thanks for your kind suggestions. This manuscript was aiming at the topic of polymer coatings with graphene oxide (and its derivates). GO (and its derivates), as a kind of nano-fillers, is an important part of composite coatings. As a result, we think that more attention should be paid to improve the performance of the coatings against corrosion with GO by protection mechanism of “labyrinth effect”.

Studies examining surface coatings and interface characteristics should be included with SEM,EDS,etc images.

-Response: Thanks for your kind suggestions. As discussed above, we agree with your opinion that SEM and EDS are indeed important in the analysis of the interface between substrate and the coatings. However, the main focus of this paper was to summarize the application of GO as filler in the coating, and the characteristics of the interface was not so crucial in the key factors (dispersion and aggregation) affecting the GO performance. In our own study of the coating performance, the SEM and EDS will be conducted to better characterize the interface performance between substrates and coatings.

The study contains very few figures and tables. studies in the literature should be given in the form of tables in comparison. The results obtained in the literature about surface coatings should be presented as figures.

-Response: Thanks for your kind suggestions. We added a table in the part of 4.2 to give a direct exhibition and comparison about the different kind of pH sensing molecules that are potential for corrosion detection. The results obtained in the literature about coatings are presented as figure 8, and we summarized the anti-corrosion performance based on the Low-frequency impedance modulus (|Z|0.01Hz) of composite coating and blank sample under optimal conditions reported in the literature listed above.

GO composites are frequently used on different substrates to increase their wear and corrosion resistance. How the coatings applied on the substrates affect the wear and corrosion resistance should be given by comparing the examples in the literature.

-Response: Thanks for your kind suggestions. To better illustrate the GO application in corrosion resistance and make comparison between different GO composite, several references have been added and compared in the discussion part of 4.3. The difference and similarity of different GO composite have been compared and summarized, and the revisions have been marked in yellow.

Round 2

Reviewer 3 Report

Thanks for answers and revisions.